# Network Contractility during Cytokinesis—From Molecular to Global Views

**DOI:** 10.3390/biom9050194

**Published:** 2019-05-18

**Authors:** Joana Leite, Daniel Sampaio Osorio, Ana Filipa Sobral, Ana Marta Silva, Ana Xavier Carvalho

**Affiliations:** 1Instituto de Investigação e Inovação em Saúde – i3S, Universidade do Porto, 4200-135 Porto, Portugal; joana.leite@i3s.up.pt (J.L.); Daniel.Osorio@ibmc.up.pt (D.S.O.); ana.sobral@i3s.up.pt (A.F.S.); amsilva@ibmc.up.pt (A.M.S.); 2Instituto de Biologia Molecular e Celular—IBMC, Universidade do Porto, 4200-135 Porto, Portugal

**Keywords:** cytokinesis, contractile ring, actomyosin contractility, modeling

## Abstract

Cytokinesis is the last stage of cell division, which partitions the mother cell into two daughter cells. It requires the assembly and constriction of a contractile ring that consists of a filamentous contractile network of actin and myosin. Network contractility depends on network architecture, level of connectivity and myosin motor activity, but how exactly is the contractile ring network organized or interconnected and how much it depends on motor activity remains unclear. Moreover, the contractile ring is not an isolated entity; rather, it is integrated into the surrounding cortex. Therefore, the mechanical properties of the cell cortex and cortical behaviors are expected to impact contractile ring functioning. Due to the complexity of the process, experimental approaches have been coupled to theoretical modeling in order to advance its global understanding. While earlier coarse-grained descriptions attempted to provide an integrated view of the process, recent models have mostly focused on understanding the behavior of an isolated contractile ring. Here we provide an overview of the organization and dynamics of the actomyosin network during cytokinesis and discuss existing theoretical models in light of cortical behaviors and experimental evidence from several systems. Our view on what is missing in current models and should be tested in the future is provided.

## 1. Cytokinesis Overview

Cell division is the fundamental process during which a mother cell partitions its duplicated genetic material and cytoplasm between the two daughter cells. The completion of cell division requires the formation of an equatorial furrow, whose continued ingression towards the center of the cell generates a physical barrier between the two daughter cells. This process is designated cytokinesis [1]. Failure of cytokinesis compromises the whole process of cell division leading to polyploid cells that may serve as a starting point for aneuploidy and tumorigenesis (reviewed in [2]). In animal cells, cytokinesis involves the formation of a contractile actomyosin ring (CR) that sits at the base of the ingressing equatorial furrow. Constriction of the CR drives continuous membrane furrowing until only a narrow cytoplasmic bridge separates the two cells. At this point, daughter cells are separated by the process of abscission (reviewed in [3]). The CR is a transient structure that forms shortly after anaphase onset and disassembles as it constricts. The assembly timing and localization of the CR are dependent on the mitotic spindle [4] and, in animal cells, are determined by the activation of the small GTPase RhoA [5,6]. RhoA is activated by guanine nucleotide exchange factors (GEFs) and inactivated by GTPase activating proteins (GAPs), the two protein families regulating the guanosine triphosphate (GTP)- guanosine diphosphate (GDP) nucleotide binding state of small GTPases (reviewed in [7]). In animal cells, the main cytokinesis RhoA GEFs are Ect2, GEF-H1 and MyoGEF, whereas the main GAPs are MgcRacGap, p190RhoGAP and MP-GAP (reviewed in [8,9]). Active GTP-bound RhoA directs the two main activities required for CR function: (1) elongation of non-branched actin filaments through the direct activation of formins [10], and (2) myosin contractility through activation of Rho-associated protein kinase (ROK or ROCK) (reviewed in [11]). The basic toolset of proteins that constitute the CR is common to many organisms throughout evolution (reviewed in [12]), but quantitative details on basic CR constituents and on their organization are still lacking for most studied systems. The CR has been more extensively studied in the fission yeast *Schizosaccharomyces pombe*, where deletion strains, conditional mutants and endogenous tagged versions of many cytokinesis genes are available. These, together with the yeast’s small size, its constant shape due to the presence of a cell wall, and the slow pace of cytokinesis, have allowed for unique collective knowledge that includes a clear timeline of events, quantitative analysis of protein levels and high-resolution description of the distribution of key intervening components [13,14,15,16,17]. 

How the CR functions in animal cells is less well understood. High-resolution structural and molecular characterization of the CR in animal cells is still lagging compared to yeast and has been the focus of recent studies [18,19]. Translation of findings from yeast to other systems has allowed progress in the field; however, clear differences between cytokinesis in yeast and animal cells exist and should be taken into consideration. Ring constriction in *Schizosaccharomyces pombe* is much slower (on average ~20 times slower than animals), occurs over a significantly smaller distance (on average ~12 times smaller than in animals), and is coupled to the synthesis of cell wall material (reviewed in [20]). Importantly, removal of all F-actin structures via latrunculin A treatment (see Table 1) after septation initiation does not preclude cell cleavage in *Schizosaccharomyces pombe* [21]. Likewise, *Saccharomyces cerevisiae* myosin deletion mutants (myo1) are able to complete septation and cell separation despite lacking a detectable actin ring [22]. This indicates that the actin ring is not essential for cytokinesis in budding and fission yeast after septation has been initiated. Another important difference likely to substantially impact CR behavior and functioning is that fission yeast lacks an actomyosin cortex surrounding the ring, whereas in animal cells, the CR is integrated in the actomyosin cortex that underlies the plasma membrane [19,23]. Overall, we have a comprehensive knowledge of CR components and their individual roles and properties. Nonetheless, our understanding of how these components work together to promote CR constriction, leading to physical separation of dividing animal cells, is still limited [24]. Mathematical integration of experimental data has provided valuable insight into the complexity of the process. In the following sections, we start by reviewing the current knowledge and working hypotheses concerning structural organization, dynamics and contractility mechanisms of the CR, with a particular focus on animal cells. We then describe the mathematical models that have been proposed on the mechanics of cytokinesis, their main outcomes and limitations. Finally, we discuss important gaps in the field that need to be addressed for a global view of cytokinesis to be achieved.

## 2. Actin and Myosin—Main Contractile Ring Components 

### 2.1. Actin Filaments

Actin is the main component of the CR and the absolute dependence of cytokinesis on actin has been reported since the early 1970s using actin depolymerizing drugs [25,26,27,28,29] (see Table 1). Filamentous actin (F-actin) is the result of polymerization of globular actin monomers (G-actin). Actin filaments have a polar structure since monomers are always integrated in the same orientation, generating two different filament ends, designated barbed and pointed ends. This polarity impacts filament polymerization dynamics as in equilibrium, monomer addition to an existing filament is ten times faster at the barbed end (therefore also designated plus-end) than at the pointed end (minus-end) where disassembly mostly happens. Within the cell, actin dynamics is modulated by many actin binding proteins that control nucleation, elongation, branching, capping or severing of filaments (reviewed in [30,31,32]). Linear actin filaments in the CR are nucleated and/or elongated by formins (reviewed in [33]), whereas actin depolymerizing factor (ADF)/cofilin plays an important role in filament severing and depolymerization (reviewed in [34]). Branched actin filaments nucleated by the ARP2/3 complex also compose the cell cortex during cytokinesis. The ARP2/3 complex prevents an excess of formin activity at the cell cortex [35] and does not localize in the CR, likely to avoid interference with formin activity (reviewed in [9,24]). 

### 2.2. Myosin Filaments

Non-muscle myosin II (hereafter myosin) is a barbed-end directed actin motor of the same family of the muscle myosins that are responsible for muscle contraction (reviewed in [36]). Myosin is an hexameric protein composed of two heavy chains and two copies of two different light chains: the essential light chain (ELC) and the regulatory light chain (RLC). The heavy chain consists of a N-terminal globular head with ATPase activity and actin-binding capacity, a lever arm where the myosin light chains bind and a coiled-coil rod in the C-terminal half. Myosin molecules oligomerize in anti-parallel fashion via the rod domains, progressing from intermediate oligomers into thick bipolar filaments with the head domains located at both ends of the filaments [37,38,39,40]. Phosphorylation of conserved serine/threonine residues in the N-terminus of the RLC are thought to cause the heavy chain to open into the extended active conformation that is competent to form bipolar filaments [41,42]. Bipolar filaments are 200–300 nm wide [18,19,43], include between 16 and 40 myosin molecules [43,44,45], and are believed to be the contractility-competent form of myosin. As myosin filaments bind actin filaments via their head domains, the same myosin filament can crosslink multiple actin filaments. In addition, myosin can drive actin filament sliding via conformational changes within the myosin heads. These conformational changes are induced by ATP hydrolysis in the nucleotide pocket and are communicated to the lever arm generating a power stroke that causes myosin to move on the actin filament (reviewed in [46]). Myosin bipolar filaments slide antiparallel actin filaments by moving towards their barbed ends, resulting in actin filament network contraction. The understanding of myosin bipolar filaments behavior is progressively changing as myosins from other families have been shown to co-polymerize with non-muscle myosin II [47].

The absolute dependence of cytokinesis on myosin has been documented for several decades, from pioneering experiments involving the injection of myosin antibodies in starfish embryos [48,49] or the disruption of the myosin heavy chain in *Dictyostelium discoideum* [50] to more recent experiments using antisense RNA injection [51] or a specific small molecule inhibitor called Blebbistatin [52].

## 3. Actomyosin Contractility in the Contractile Ring

### 3.1. Actomyosin Structure of the Contractile Ring

Muscle sarcomeres are perhaps the most well-characterized actomyosin structure. Skeletal muscle is organized in myofibrils composed of series of sarcomeres. Each sarcomere is limited by two Z-discs, where parallel actin filaments are anchored by their barbed ends. Actin filament pointed ends point towards the center of the sarcomere. Actin filaments are intercalated with myosin bipolar filaments that are anchored in the M-band at the middle of the sarcomere. As myosin heads move towards the barbed ends of the actin filaments, the sarcomere constricts to a limited extent determined by the maximum overlap between actin and myosin filaments. Not surprisingly earlier studies of CR contractility evaluated the possibility of sarcomere-like constriction [27,53]. However, clear sarcomeric modules have not been observed by electron microscopy (EM) in the ring. In addition, contrary to skeletal muscle, the CR maintains its thickness as it constricts, which implies that actin filaments disassemble as its circumference decreases.

Seminal EM studies by Schroeder in the 1970s demonstrated the filamentous nature of the CR structure in sea urchin or jelly fish eggs and human cultured HeLa cells [27,54,55]. The CR was shown to be positioned just below the plasma membrane of the cleavage furrow and presented closely packed parallel actin filaments. The width of the CR was 3–17 µm in sea urchin eggs, 6 µm in jelly fish eggs, and 10 µm in HeLa cells, whereas its thickness was about 0.1–0.2 µm for the different cell types. The actin filaments of the CR were shown to bind heavy meromyosin (HMM) [55] and the improvement of fixation methods allowed to use HMM-labelling to reveal the existence of opposite polarities of actin filaments within the CR [53]. The presence of myosin filaments in the CR was first documented by fluorescence microscopy of cultured cells using myosin specific antibodies [48,56], and then reported at the ultrastructural level with the visualization of myosin-like filaments alternating with actin filaments and parallel to the direction of constriction [53,57]. Additionally, the furrow membrane presented underlying round patches of electron dense material that appeared to be associated with the ends of actin filaments, thus constituting potential anchor sites for the CR at the plasma membrane [57]. A recent study using platinum replica EM of isolated cortices of dividing sea urchin embryos confirmed the overall organization reported in earlier studies [19]. This study showed that the CR is a specialized structure integrated within a continuous actomyosin cell cortex. Actin filaments within the CR were primarily organized into compacted non-branched bundles, aligned parallel to the cleavage axis. The surrounding cortex, on the other hand, mainly consisted of a crisscross F-actin meshwork [19,35,58,59] (Figure 1). Unlike other organisms, *D. discoideum* contractile rings were shown to be a disorganized meshwork of actin filaments throughout cytokinesis, as observed by platinum-shadowed transmission EM, platinum-shadowed scanning EM and three-dimensional (3D)-EM tomography [60]. Whether the actin filaments that form the ring are recruited from adjacent regions of the cortex or assembled de novo at the cell equator during CR assembly in animal cells is still unclear. In support of the first, it has recently been proposed that actin filaments flow along the cell cortex towards the cell equator, and that compression due to persistent unidirectional flows towards the equator of the cell is the main driver for actin filament alignment at the equator and furrow formation in one-cell *Caenorhabditis elegans* embryos [61]. In support of the second, cytokinetic formins responsible to nucleate and elongate the actin filaments in the ring are activated at the cell equator where Rho-GTP localizes [62,63,64,65] (Figure 1). 

Myosin filaments in the CR are organized in broadly distributed clusters in early stages of cytokinesis and in tight, linearly aligned bundles oriented parallel to the cleavage plane in later stages as revealed by 3D-structured illumination microscopy of isolated cortices of dividing sea urchin embryos. The orientation of myosins in the early clusters was difficult to resolve but imaging of later stages of cytokinesis suggested the presence of head-to-head arranged myosin filaments that associated laterally [19]. Recent super-resolution microscopy with differential tagging of N- or C-terminal regions of myosin filaments shed light into the organization of myosin molecules in the CR in HeLa cells. HeLa cells express Myosin IIA and B, and Myosin IIA filaments were found to group in supramolecular structures designated as stacks [18], also observed in stress fibers [66]. Myosin IIA stacks oriented parallel to the cleavage furrow plane, an optimal orientation for the sliding of actin filaments. The formation of stacks was impaired in the presence of low concentrations of Blebbistatin, when a motor impaired version of myosin was expressed or after cytochalasin B treatment, indicating that both myosin motor activity and actin filaments are important for the formation of these myosin suprastructures that may constitute contractile units within the CR [18]. Whether myosin filaments form at the cell equator or are transported from elsewhere is unclear. Myosin may be recruited to the equatorial region by a diffusion-equatorial retention mechanism, as described in cultured cells [67,68,69]. According to this hypothesis, myosin would diffuse from the cytoplasm and become retained at the equator as a result of filament formation, which is promoted by the phosphorylation of its regulatory light chain by kinases that are locally activated by Rho-GTP [5,68,70]. Myosin also flows from the cell poles to the cell equator in *C. elegans* 1-cell embryos and was proposed to enter the equatorial Rho-GTP zone, where it is activated and loaded into the ring [71]. A mechanosensing-mediated mechanism inherent to myosin motors has also been proposed to lead to cooperative myosin accumulation at the furrowing cortex [72]. 

How the organization of the CR progresses throughout cytokinesis is still poorly characterized. In fission yeast, rings form from nodes or protein clusters of Mid1p (anillin-like protein), Cdc15p (F-BAR protein), Rng2p (IQGAP), and Cdc12p (formin), that anchor myosin Myo2p to the plasma membrane and extend actin filaments from their barbed ends [15,73]. Anchored myosin in neighboring nodes grabs extending actin filaments and pulls on them, allowing the nodes to coalesce and form the ring [15,73,74,75,76,77]. As the process progresses, the randomly growing actin filaments align along the ring [75]. Work in cultured vertebrate cells also suggested that the CR starts with randomly oriented F-actin filaments that progressively orient parallel to the cleavage furrow due to a substantial increase in mechanical tension along the cell equator [69]. 

### 3.2. Ring Constriction Driven by Myosin Motor Activity

The available structural data is compatible with a scenario of myosin motor activity driving contractility by sliding antiparallel actin filaments. Motor-dependent contractility can occur in the context of parallel F-actin networks organized in a sequence of units or in a continuous bundle with actin filaments in random orientations (reviewed in [78,79]). 

Although a modular structure has been proposed to exist in the CR, this kind of organization has not been observed in EM studies so far. In fission yeast, the organization in nodes persists throughout cytokinesis despite the fact that Mid1p leaves the nodes before constriction initiates. However, their positions change indicating lateral mobility on the plasma membrane [15,80]. Lateral mobility of myosin was also reported to occur in *C. elegans* constricting rings, which have been proposed to organize in a series of equal-sized contractile units in order to explain similar ring constriction times for rings with different initial sizes [81]. The observation of clusters of myosin and other ring components was also documented in mammalian cells [80]. These results point to the idea that at least some contractile ring components may be organized into discrete functional units throughout constriction. Whether the number of units remains constant during constriction is unclear and will be important to clarify for a mechanistic understanding of constriction. 

In the context of a continuous F-actin bundle lacking modularity, myosin motor activity can also lead to contraction by actin filament buckling or polarity sorting. The former stems from the intrinsic properties of actin filaments since they can support large tensile stresses but buckle readily when under compression (reviewed in [82]). In a continuous structure of actin filaments with random orientations, these intrinsic properties lead to net contraction of the network. Actin filament buckling has been proposed to be caused by the action of myosin filaments with slightly different velocities [83]. Filament buckles have higher curvature and should therefore be more prone to severing [84,85]. Whether a buckling mechanism is functionally relevant in vivo, where networks with short actin filaments and compact mesh sizes seem to exist, remains unclear [19,86,87]. The alternative mechanism is based on myosin-driven polarity sorting, which requires the observed capacity of myosin to dwell at the end of several actin filaments after processively running along them [88,89,90]. This activity leads to the formation of actin asters that can act as contractile nodes, which drive contraction in crosslinked networks with dense mesh sizes.

All these mechanisms are dependent on myosin motor activity. Although myosin motor-driven constriction is traditionally thought to be the mechanism that drives furrow ingression, experimental data that directly shows the requirement of myosin motor activity for animal cytokinesis in vivo is limited. Most studies have used the small molecule inhibitor Blebbistatin, the depletion/inactivation of myosin or myosin temperature-sensitive mutants, as well as non-phosphorylatable mutants of the regulatory light chain, to draw specific conclusions about the requirement of myosin motor activity during cytokinesis [52,58,61,91]. However, all of these approaches are limited in distinguishing the motor and F-actin crosslinking contributions of myosin (see Section 3.3). The myosin inhibitor Blebbistatin stabilizes the transition state of the molecule where the 50 kDa cleft where actin binds is only partially closed. Therefore, although Blebbistatin precludes the force-generating steps, it also keeps myosin in a low actin affinity state [92]. Likewise, neither the depletion of myosin nor the inactivation of myosin temperature-sensitive mutants allow the differentiation between motor and non-motor capacities of myosin, and interfering with the regulatory light chain may affect other myosins or perturb myosin localization and/or structure [93,94,95] (see Section 2.2). To investigate the requirement for motor activity specifically, work in different systems addressed the effect of expressing different point mutations in the myosin head that impacted motor activity, according to biochemical characterization. In fission yeast, the expression of a myo2 mutant version containing three point mutations (G345R, Q640H and F641I) that bound tightly to actin and was unable to slide actin in vitro was sufficient for CR assembly but not constriction, whereas the mutant containing just the G345R mutation that did not attach to actin did not allow for CR assembly [96]. *Dictyostelium discoideum* myosin-null cells were able to perform cytokinesis when attached to a surface, but grew very slowly when in suspension [50,97]. In agreement with this, expression of motor impaired mutants also showed differential behaviors between adherent cells and cells grown in suspension. For example, expression of a myosin mutant that bound tightly to actin but was motor-dead (MYS2(S237A) [97]), led to slow growth in suspension and multinucleation in adherent cells. However, expression of slower myosins (MYS2(S236A) [97] or MYS2(S456L) [60]) or a motor-dead myosin that seemed to bind weakly to actin (MYS2(R238A) [97]) strongly affected cell growth in suspension but allowed for successful cytokinesis in adherent cells. As for animal systems, adherent COS-7 cells (that lack myosin IIA and IIC) expressing supposedly motor-dead myosin IIB R709C, N93K, or R234A, all with actin-binding capacity were able to complete cytokinesis after RNAi depletion of endogenous myosin IIB [98]. In contrast, recent work from our lab in *C. elegans* zygotes showed that neither expression of myosin NMY-2(R252A) mutant (corresponding to *D. discoideum* MYS2(R238A) and myosin IIB(R234A)) that is motor-dead and has weaker actin binding capacity than wild-type, nor NMY-2(S251A) mutant (corresponding to *D. discoideum* MYS2(S237A) [97]) that is motor-dead and binds tightly to actin, were able to support cytokinesis [99]. It is noteworthy that two of the three mutants used in the above-mentioned COS-7 cell study [98], myosin IIB(R709C) and myosin IIA(N93K), should not be considered bona-fide motor-dead versions of myosin: the mutant myosin IIA(N93K) has recently been clarified by the Adelstein and Sellers laboratories not to be a motor-dead myosin [100], and the corresponding mutation to myosin IIB R709C in *C. elegans*, NMY-2(R718C), only partially affects motor activity [99], which is in agreement with the corresponding mutation in mammalian myosin IIA(R702A) that was reported to translocate actin filaments at 50% the velocity of the wild-type counterpart [101]. Expression of the third mutant R234A, which consistently stands as a motor-dead mutant in *D. discoideum*, mammalian cells [100] and *C. elegans*, is able to ameliorate cytokinesis defects in COS-7 cells depleted of myosin IIB and allow for cytokinesis in attached *D. discoideum* cells, while it does not support cytokinesis in *C. elegans* embryos or in *D. discoideum* growing in suspension. These results could indicate that cytokinesis has a differential requirement for myosin motor activity in these systems or in particular conditions when cells are adherent to a substrate (see next section). In agreement, some adherent cultured cells can divide with a compromised contractile ring in an adhesion-dependent manner by migrating away in opposite directions during cell division [102,103,104]. Additionally, contribution of other myosins should be considered when interpreting and comparing results from different systems.

### 3.3. Ring Constriction Independently of Myosin Motor Activity

A possible constriction mechanism envisioned for the CR is based on F-actin dynamics and the crosslinking abilities of both myosin and other actin-binding proteins (see more on Section 3.5). Indeed, actin filament polymerization and depolymerization can generate stress if in the presence of end-tracking crosslinkers, i.e., crosslinkers that remain bound to depolymerizing filament ends. Checking for this possibility in vivo is challenging, because it involves (1) expressing myosin mutants that do not translocate actin but still bind to it, preferably in a tight fashion to guarantee crosslinking; and (2) in case cytokinesis does not fail after expression of myosin mutants, combining with depletion of actin crosslinkers. Although myosin motor is known to be important for ring constriction in *C. elegans* embryos [99] and in *S. pombe* [96], its requirement has been contested in mammalian cells [98], in *D. discoideum* growing on a surface and in *S. cerevisiae* [105,106,107] (see Section 3.2). As to non-motor end-tracking crosslinkers, none have been identified yet, but more investigation is required. To distinguish the contribution of myosin motor activity and F-actin turnover is a hard task, since myosin is known to be implicated in F-actin disassembly [84,85,107,108,109]. Many studies, however, consider that a combination of F-actin turnover or disassembly and myosin-driven contractility contribute to overall ring contraction [75,81,107,110].

Myosin motor-activity independent cytokinesis in adherent *D. discoideum* cells has been proposed to result from a combination of crawling forces that help cells elongate and a Laplace pressure generated by material properties that facilitates furrow progression [111]. In animal cells, adhesion to the substrate has also been shown to allow for cell division to occur when the CR is compromised [104].

### 3.4. Importance of F-Actin Dynamics for Ring Constriction

Actin filament turnover through continuous polymerization and depolymerization ensures the maintenance of pools of F-actin and G-actin for constant remodeling of the actin cytoskeleton. Both polymerization and depolymerization of F-actin are thought to occur during ring constriction. To understand the contribution of actin dynamics specifically during ring constriction, researchers capitalize on the use of acute treatments with actin-interfering drugs, whose origin and overall effects on F-actin networks we summarize in Table 1

Addition of latrunculin or cytochalasin to mammalian LLCPK1 cells, sea urchin eggs or fission yeast during ring constriction leads to cytokinesis failure [21,27,80,119,120]. In early *C. elegans* embryos, addition of latrunculin A after furrow initiation was reported to slow ring constriction [81]. With the advance of a reliable assay for embryo permeabilization and acute drug treatments in this system [121], we have now extended these experiments. Using this assay, we confirmed that low doses of latrunculin A slow down ring constriction [122], and higher doses cause a single break in a random location of the ring circumference, precluding cytokinesis completion (Figure 2). 

Injection of cytochalasin D or latrunculin A in the cytoplasm of normal rat kidney (NRK) cells, close to the cytokinetic furrow, led to disassembly of the equatorial actin cortex and facilitated furrowing. However, the same localized drug treatment at the polar cortex inhibited cytokinesis progression, indicating that polar and equatorial cortical actin polymerization may be governed differently to coordinate CR closure [123]. Addition of jasplakinolide or a mix of jasplakinolide and latrunculin (to prevent jasplakinolide-induced ectopic actin polymerization) led to ring constriction slowdown in yeast [107,120] and arrested cytokinesis in LLCPK1 cells [119]. Although the results are not consistent in all systems tested, overall these experiments support the idea that actin filament dynamics are required for cytokinesis. Further insight will certainly be obtained when better spatio-temporal control over F-actin dynamics, using for example photo-caged drugs, becomes available to test in a variety of systems. 

Besides drug treatments, there is additional evidence supporting the importance of actin filament polymerization and depolymerization during ring constriction. A robust mechanism to repair potential discontinuities in the ring structure was found to exist during constriction and to depend on actin polymerization [122]. This indicates that actin polymerization occurs during ring constriction and is required for the mechanical robustness of the CR. Additional evidence for actin polymerization being required during ring constriction should come from the understanding of formins activity after the CR is assembled. Nucleation/elongation of actin filaments by Rho-GTP-activated formins is required during CR assembly [58,124,125], but the continuous requirement of formin activity throughout ring constriction is less well explored. So far, it is known that the formin CYK-1 remains active throughout ring constriction in *C. elegans* early embryos, as its inactivation during ring constriction considerably slows it down [35]. Evidence that actin depolymerization is required for cytokinesis comes from the fact that the contractile ring does not increase in thickness [27,81] and measurement of actin filament length by transmission EM indicates that filaments become shorter during constriction [86], both indicating that net F-actin disassembly/depolymerization occurs as the ring gets smaller. Moreover, depletion of ADF/cofilin results in the accumulation of actin filaments along the equator, leading to cytokinesis failure [126,127,128,129].

Interestingly, studies using yeast cell ghosts that retained a contractile ring surrounded by a permeabilized plasma membrane and devoid of cytoplasmic structures revealed that ring constriction in vitro requires myosin-dependent contraction but does not require F-actin dynamics, namely actin depolymerization, because rings contracted at normal speed after treatment with jasplakinolide or inactivation of ADF/cofilin [130]. In addition, formaldehyde-fixed ghost rings could constrict in the presence of a processive vertebrate myosin-V that is unipolar and does not form filaments [120]. This together with experiments in intact yeast and yeast spheroplasts using latrunculin versus latrunculin and jasplakinolide indicates that actin turnover may not be required for ring contraction per se, but rather to maintain actin filament homeostasis during the process. This evidence argues against the idea that F-actin dynamics constitutes the main driving force for ring constriction and instead strengthens the possibility that it is required to maintain the structure of the CR. Nevertheless, the predicted rapid turnover of actin in *S. pombe* protoplast rings was proposed to be key for maintaining ring tension and ring remodeling during constriction [75]. In sum, further experimentation will be necessary to clarify what the contribution of actin dynamics to ring constriction is.

### 3.5. Involvement of Actin Filament Crosslinkers in Cytokinesis

Actin filament crosslinkers are proteins able to bind and bridge actin filaments. Myosin can be classified as an active crosslinker, since it is capable of bridging as well as translocating actin filaments. Non-motor or passive crosslinkers, like alpha-actinin [131], fimbrin/plastin [132], filamin (reviewed in [133]), spectrin (reviewed in [134]), anillin [135,136] or septins [135,137], are proteins that bridge actin filaments, either by promoting tight parallel bundling or loosely crosslinked F-actin networks. Depending on their size, concentration, affinity to actin filaments and structural properties, these crosslinkers lead to the formation of actin filament networks with distinct morphologies and stiffness levels. The role of non-motor crosslinkers during cytokinesis is not sufficiently explored. For instance, it is still unclear whether the simultaneous expression of several crosslinkers reflects functional redundancy or a necessary diversification. There is limited evidence that redundancy exists and that includes the fact that while fission yeast single mutants of alpha-actinin and fimbrin are viable, a double mutant is not, due to problems during cytokinesis [138]. Another open question has to do with the localization of these proteins in dividing cells. Some of the crosslinkers have been shown to localize in CRs, such as alpha-actinin [139], fimbrin/plastin [59,140,141,142], anillin [143], and septins [144], but all of them also localize in the surrounding cortex. Whether crosslinking effects on cytokinesis progression are associated with their function in one location or the other is still a subject of study.

Depletion of individual crosslinkers does not consistently lead to cytokinesis failure. For instance, alpha-actinin, one of the most popular crosslinkers for in vitro studies, was shown to be dispensable for early embryonic cytokinesis in *C. elegans* [59], and its depletion was found to accelerate furrow ingression in NRK mammalian cells [145]. To our knowledge, only depletion of cortexillins in *D. discoideum* and of anillin in *Drosophila melanogaster* S2 and human HeLa cells (but not in *C. elegans* embryos) were described to lead to significant cytokinesis failure [136,146,147,148,149]. Nevertheless, cortexillins do not seem to be well conserved in higher organisms, and anillin has additional functions, and therefore its requirements may not be due to its F-actin crosslinking activity. Overexpression of crosslinkers seems to more consistently lead to multinucleation in a variety of systems [60,138,145,149,150]; however, the effect of overexpressing crosslinkers is not much revealing in what concerns the normal function of these proteins. Non-motor crosslinkers seem to contribute to early cytokinesis. In *C. elegans* embryos, fimbrin/plastin loss of function led to reduced actin filament connectivity and impaired flows towards the cell equator but did not affect ring constriction rate [59]. A combination of in vivo experiments and computational modeling in fission yeast suggested that alpha-actinin and fimbrin cooperate with myosin during the initial self-organization of the ring [150], and alpha-actinin turnover is necessary to better organize ring components and facilitate node coalescence [151]. The mobility/dynamics of the crosslinkers dynacortin, fimbrin, and primarily cortexillin was found to be decreased at the cleavage furrow when compared to the polar regions [60] and interphase cortex of *D. discoideum* [152]. This decrease could be due to increased mechanical stress at the furrow region. Cooperative accumulation of myosin and cortexillin I was observed during micropipette aspiration of polar regions during cytokinesis in *D. discoideum* [72,153]. As a result, a mechanosensory response mediated by myosin and cortexillin I was suggested to monitor cell shape and correct defects through the recruitment of myosin and cortexillin to the site of cell deformation in this system [72,153]. The cooperativity between myosin and cortexillin I was proposed to result from myosin inducing or stabilizing conformational changes of the actin filaments, which in turn potentiate enhanced binding of myosin itself and cortexillin I [154,155].

### 3.6. Modulation of Actomyosin Contractility by Actin Filament Crosslinkers 

The joint action of myosin and non-motor crosslinkers on F-actin regulates the dynamic properties of the cytoskeleton and the overall mechanical properties of the cell cortex. Indeed, myosin pulling on crosslinked F-actin networks leads to the generation of tension and its propagation, allowing networks to deform as a whole. In silico models and in vitro data indicate that non-motor crosslinkers are critical to interconnect filaments into contracting networks [91,156]. The effects of actin network architecture in contractile ring dynamics have been studied using in vitro minimal systems consisting of micropatterned rings of purified components that are able to contract in the presence of myosin [109,156] and display certain properties common to their in vivo counterparts, like the scalability of constriction rate with ring size [81,109]. These studies used a double headed HMM version of myosin VI that allows for more consistent contractility, as it does not require filament formation for processivity. However, it is noteworthy that myosin VI is a minus-end directed motor, in contrast to non-muscle myosin II. In vitro micropatterned actin rings made of ordered anti-parallel filaments contracted faster than those made of branched filaments, and addition of alpha-actinin slowed down both kinds of rings. Rings composed of parallel filaments of mixed polarity, only contracted in the presence of alpha-actinin. The same crosslinker was therefore found to increase or decrease contractility depending on the F-actin network architecture. Theoretical considerations concerning network interconnectivity and contractility revealed that networks with either low or high interconnectivity were less contractile, while intermediately interconnected networks presented optimal contractility. Other in silico simulations supported this conclusion [59,91,151]. Overall, these studies show that the contraction of an actomyosin filament network is determined by its global architecture and degree of interconnectivity. Nonetheless, whether ordered/modular anti-parallel filaments or disordered mixed-polarity parallel filaments best reflect the cytokinetic CR remains unclear and is for now debatable (see Section 3.1 and Section 3.2). In addition, the architecture of the cytokinetic ring may change throughout constriction and these approaches do not replicate actomyosin dynamics as observed in a living cell. Interestingly, in vivo studies using *C. elegans* early embryos partially depleted of anillin indicated that its normal levels limit the rate of ring constriction, as partial depletions led to an increase in the rate. Likewise, intermediate levels of myosin (NMY-2) resulted in higher constriction rate than that of non-depleted rings, whereas more severe depletions decreased the rate [91]. These studies suggested that the normal amount of myosin and crosslinking proteins does not allow for maximum speed of ring closure, but it will be important to further address this issue in vivo, for other crosslinkers and in other systems. The reason cytokinesis would not be optimized for maximal speed is enigmatic but it is likely that faster ring constriction comes with a price for the cell: since crosslinkers are also located outside the contractile ring, faster constriction as a result of decreased levels of crosslinkers could lead to decreased cortical tension, and this could compromise the integrity of the cell.

In sum, it is becoming clear that the combined action of motor and non-motor F-actin crosslinkers regulate network stiffness, tension, interconnectivity and contractility during cytokinesis. However, mechanistic insights linking regulatory pathways to the mechanical changes that take place, information about all crosslinkers involved, their number and location in the ring versus surrounding cortex, distribution of myosin activity, as well as a detailed description of F-actin organization are still missing in animal cells. This collective knowledge will be necessary for the global understanding of cytokinesis biomechanics.

## 4. Theoretical Modeling of Cytokinesis

The field of mechanobiology focuses on understanding how cell mechanics and physical forces act as dynamic players in biological phenomena and how they translate into biochemical responses through mechanotransduction. Physics began to percolate into the cytokinesis research as an attempt to explain the dramatic deformation and shape changes occurring during the process. The contribution of mathematical modeling has been and will continue to be fundamental to increase our global understanding of this complex process.

### 4.1. Modeling of Actomyosin Networks 

Theoretical modeling of biological systems consists of developing algorithms, data structures, and other tools that can simulate components, as well as interactions and interplays between them, and predicting outcomes in different biological contexts. It has the power to predict the behavior of complex biological systems by greatly simplifying the molecular details, keeping only the essential variables and reducing the number of parameters to be considered. Currently, two general types of computational models can be identified in the study of actomyosin networks: agent-based and coarse-grained continuum models. Agent-based modeling requires detailed information on the components and mechanisms at the molecular level, including number of molecules, filaments or molecular ensembles. By contrast, coarse-grained continuum models do not require detailed knowledge of the interactions between specific components, but rather describe the network as a continuous macroscopic material. Each type of model has its own advantages and limitations and, as such, the model chosen to address a given biological problem will always depend on the question itself and the level of detail required to answer it. While agent-based models are generally attractive as proteins’ numbers and properties can be individually played with, their main limitation arises from the difficulty in knowing exactly how each component contributes to set the network mechanical properties. Moreover, numerical simulations may become computationally unworkable when modeling large-scale systems such as the entire cell cortex during cytokinesis. This may be simplified by simulating only a portion of the CR and a portion of the rest of the cortex. Still, coarse-grained approaches may be more adequate to describe the mechanics of the whole cortex, especially when it comes to relate mechanics at the equator with mechanics at the poles and may prove particularly suitable when the biological system is poorly characterized biochemically. Nevertheless, these types of models do not provide the mechanistic understanding of the mechanical properties arising from molecular processes that often biologists are looking for. Additionally, they still require the input of chemical constants and coefficients, which need to be measured experimentally or matched to experimental results. A great example of this approach is the active gel theory [157,158,159] that has been successful in predicting cell cortex behavior in several biological processes such as cell migration [160] and cytokinesis [110,161]. In short, the active gel theory aims to apply general hydrodynamic equations to biological systems composed of cytoskeletal filaments and motor proteins that present gel-like behavior. This description retains the minimal, yet essential, properties of the cytoskeleton: dynamic polar filaments and motors with network contracting ability [157], whose equations are similar to the ones used for non-biological systems. 

### 4.2. Modeling Cytokinesis

How to mechanically explain the dramatic deformations occurring during cytokinesis has been a long-standing question. The first theoretical models of cytokinesis were inspired by pivotal experiments of cleavage in animal cells that showed that division occurs through a differential distribution of stresses and forces over the cell surface, with the tension at the equator being higher than the tension at the poles [162,163,164,165,166,167,168,169]. One of the first theories by Mitchison and Swann—the expanding membrane theory—proposed that cytokinetic furrow formation and ingression was a passive event resulting from active expansion at the poles [163,170,171]. In opposition, Marsland postulated that the furrow region was itself contractile, independently of the poles [172,173,174]. Wolpert put forward the astral relaxation theory, which posits that relaxation at the poles is a result of signaling cues from the astral microtubules, thus allowing the furrow region to contract [175]. The first mathematical models proposed between the late 1970s and the 1990s were purely theoretical and consisted of continuous descriptions of the entire cell surface [176,177,178,179,180,181]. White and Borisy [178] developed the first purely computational model of cytokinesis, capitalizing on the astral relaxation theory [175]. They suggested that the development of cortical tension anisotropy drives furrow assembly and ingression. In their model, the cortex was considered a random network of linear elements evenly distributed, where elements between the poles and the equator moved towards the equator due to a laterally directed force arising from the gradient of cortical tension. This would drive the accumulation of contractile elements at the equator, leading to higher cortical tension in this region. More recent computational models of cytokinesis have mostly focused on understanding the behavior of the CR during the constriction phase, generally disregarding the contribution of the surrounding cortex in setting constriction dynamics. These models are either coarse-grained [161,182,183,184] or agent-based microscopic descriptions that mostly focus on interactions between actin filaments and myosin motors [75,91,107,185,186] and will be discussed in more detail below (Figure 3).

Description of variables used in the equations. a) Agent-based models. Hooke’s Law: k—stiffness; L—length; L0—resting length. Euler Buckling: κ—stiffness; K—effective length factor accounting for the boundary conditions applied on a filament; L—critical length. Stokes-Einstein Relation: D—diffusion constant; K0—Boltzmann’s constant; T—absolute temperature; η- viscosity; r—radius. Binding Kinetics: Ka—binding constant; kon and koff—rates of association and dissociation, respectively; [A] and [B]—concentrations of molecular components. Probability of events (Gaussian Distribution): P—probability of events; x- random *variable;* µ- mean; σ2—variance. b) Coarse-grained models. Laplace’s Law: P—pressure; T—tension; R—radius. Cauchy momentum equation: ρ- density; D/Dt—material derivative; u—flow velocity; ▽—*divergence*; p—pressure; t—time; τ- stress tensor; g—accelerations. Nematic orientation: Q(S,t)—nematic orientation parameter; α(S,t)—orientation angle at position S and time t. Reaction-Diffusion: q—velocity vector; D—diagonal matrix of diffusion coefficients; ▽2—Laplacian operator; R(q)—all local reactions. 

### 4.3. Models of Contractile Ring Constriction

Ring constriction has been a subject of extensive mathematical modeling. Actin filaments, as well as myosin motors and non-motor crosslinkers (see Section 3), are common components to all mathematical models of ring constriction. As actin filament architecture in the CR is still not clear (see Section 3.1), most models assume that the ring is simply a region with higher degree of actin filament alignment and higher myosin concentration. Models are starting to include the ideas that motor- and non-motor mechanisms can generate contractility (see Section 3.2 and Section 3.3) and network dynamics, as molecules are constantly turning over, which defines characteristic timescales for interactions and mechanical properties. 

Carlsson et al. evaluated the force generated by myosin filaments in actin gels and estimated that network contraction by motors does not generate the necessary force for ring constriction. For that to happen, actin filaments must be crosslinked into long units, which would compensate for the small size of individual actin filaments [187]. Zumdieck et al. suggested that, in addition to myosin motors, contractile stress can also be generated by actin depolymerization coupled to the action of end-tracking crosslinkers [186]. In their microscopic model, myosin motors produce stress in the network by sliding actin filaments, which can polymerize and depolymerize. These molecular details were then coarse-grained into a continuum description, where polar actin filaments could change orientation due to diffusion, as well as polymerization and depolymerization, and the roles of myosin and end-tracking crosslinkers were represented by coefficients of tension contribution. Finally, these coefficients were interpreted in terms of a phenomenological model that considers the stress generated in filament bundles, and how this results in constriction. Following up on this study, another microscopic model was able to recapitulate part of ring constriction in budding yeast by including actin filaments, end-tracking crosslinkers and myosin motors to compute macroscopic parameters describing constriction dynamics. Actin filaments in the ring were allowed to overlap, change orientation and length, but effects of actin polymerization during constriction were not considered. Also, actin filament sliding by myosin and binding of crosslinkers were not allowed to occur simultaneously for the same filament. This model provided evidence for a major role of actin disassembly in driving ring constriction [107]. Of note, none of these agent-based models considered the likely contribution of membrane-anchoring of the barbed-ends of actin filaments via formins and/or myosin to the generation of tension. 

A recent study in the *C. elegans* embryo addressed the topic of network crosslinking and the idea that wild-type rings are not optimized for maximum constriction speed (see Section 3.6). The authors used an agent-based model to simulate isolated CRs consisting of actin filaments, myosin motor ensembles and actin crosslinkers, whose amounts were extrapolated from fission yeast and scaled to *C. elegans* rings. They proposed a mechanism in which intermediate concentrations of motor and non-motor crosslinkers prompt faster constriction, which is attenuated by higher or lower concentrations of these components [91]. 

The first evidence that the CR generates tension dates back to Rappaport experiments in sea-urchin embryos with glass microneedles. Rappaport showed that cleavage furrows of dividing embryos produce tensions in the range of 8–16 nN, based on their ability to bend the microneedles [166,188]. However, how the ring organizes and belts up to generate this tension is still a subject of debate (see Section 3.1). In an attempt to uncover the mechanisms through which the ring produces tension, Stachowiak et al. proposed an agent-based model of the *S. pombe* CR including experimentally measured amounts of actin filaments, actin-elongating formins, myosin and the crosslinker alpha-actinin, as well as rates of actin filament elongation, turnover of myosin and formins, force generated by myosin motors and drag resulting from membrane-anchors and crosslinkers [75]. This CR model, driven by myosin-dependent actin filament sliding and uniformly distributed anchors connecting the ring to the membrane, reliably reproduced the organization, dynamics and tension observed experimentally in rings from fission yeast protoplasts [189]. Importantly, the authors show that effects of turnover, not only for actin filaments, but also for other ring proteins, are essential to renew ring organization and sustain force, thus avoiding the accumulation of myosin into clusters and consequent loss of tension.

To reproduce ring closure time and asymmetric furrowing in *C. elegans* zygotes, a coarse-grained description of the CR used energetic parameters associated with progressive membrane curvature-dependent F-actin alignment and force generation by myosin. In this model, the CR was considered to be a membrane-bound series of contractile units connected by spring-like connections mimicking actin crosslinkers. Unit constriction was achieved by myosin-dependent actin filament sliding, where actin filaments on each unit had a random orientation initially and progressively aligned as the ring constricted. CR constriction was driven by a positive feedback mechanism where membrane curvature promoted actin filament alignment and consequently contractility, which in turn increased membrane curvature [184].

Recently, a 3D microscopic model based on electron cryotomography data [17,190] suggested that constriction in fission yeast CRs is achieved by an interplay between two distinct myosin structures—bipolar and unipolar. Actin filaments, bipolar and unipolar myosins and actin crosslinkers were all modelled as beads connected by springs, while the membrane was modelled as a sheet of beads, initially presenting a cylindrical shape. The ATPase cycle of myosin, which dictates the binding state of myosin to actin (see Section 2.2), was also integrated, as an attempt to make actomyosin interactions more realistic. The authors tested all possible configurations of actin filaments and myosins (bipolar or unipolar), including attachment to the membrane, and concluded that their simulations support a model in which ring tension is primarily generated by sliding of actin filaments via bipolar myosins and transmitted to the membrane via unipolar myosins, which would be attached to the membrane individually [190]. However, neither bipolar nor unipolar myosin filaments could be resolved in the cryotomography data [17], and therefore, this hypothesis still lacks support from experimental data.

### 4.4. Open Questions and Future Directions on Cytokinesis Modeling

#### 4.4.1. The Influence of the Surrounding Cortex

Despite cytokinesis being inherently associated with the constriction of the CR, one should not disregard the fact that the CR is a differentiated part of the cortex in animal cells. Even though recent models have mostly focused on the CR itself, disregarding the contribution of the surrounding cortex, there is both theoretical and, to lesser extent, experimental evidence indicating that the cortex surrounding the ring impacts its behavior. One of the first theoretical models of cytokinesis, by Yoneda and Dan, considered that the polar cortices, which were approximated to two portions of a sphere under tension, resisted ring constriction. Based on the Hooke’s law and considering a uniform surface tension at the cell poles, they were able to estimate the minimal equatorial force required for furrow contraction [169]. This model was used later to test experimental observations in *D. discoideum* [111,191]. Work from the Robinson lab revealed that the small GTPase RacE and the actin crosslinker dynacortin, both enriched in polar regions, act together to slow down cytokinesis. Their modeling approach revealed that after the furrow initial deformation, once the geometry of the daughter cells changes to establish an intercellular bridge, viscoelastic compressive stresses slow down bridge thinning, which is itself accomplished through myosin-dependent contractile stress. In the absence of dynacortin/RacE, compressive stresses decrease, and furrow thinning accelerates [111]. This study proposed that resistive forces from the surrounding cortex counteract actively generated contractile tension at the cell equator. Moreover, Poirier et al. developed a computational model that pinpointed several stress-generating mechanisms contributing to furrow ingression in *D. discoideum*: myosin-mediated force generation, polar protrusive forces, viscoelasticity of the cytoskeleton, cortical tension and surface curvature. Their simulations indicated that polar protrusion forces or myosin-dependent contractile forces are sufficient to initiate furrow ingression, while passive forces from cortical tension and surface curvature contribute to complete ingression [192]. Also, in support of cortical resisting forces counteracting ring constriction, laser microsurgery in constricting CRs of *C. elegans* embryos showed that the ring snaps open immediately after laser cutting, with severed ends pointing outwards and adopting a straight conformation, implying that the ring is countering cortical tension [122]. 

Turlier et al. developed a theoretical model of the dividing sea-urchin embryo based on the active gel theory [110]. Here, the cortex was considered a thin layer generating both viscous and active tensions and subjected to uniform cytoplasmic pressure. An overactivity at the equator reflecting RhoA activation necessary for furrow initiation was simulated with a gaussian distribution imposed along the cortex and centered at the equator, according to previous experimental observations [5]. This model used the same hydrodynamic theory already developed by Salbreux et al. [161], but actin filament turnover was considered, allowing the thickness of the cortical layer to vary during constriction. Actin filament turnover was found to be key for the rate and completion of ring constriction. A scaling model, in light of previous work from White and Borisy [178] and Bray and White [193], successfully explained and reproduced the properties of furrow ingression and demonstrated that the contractility difference between the poles and the furrow region was the key parameter for both constriction dynamics and cytokinesis completion.

Interestingly, a recent study has hypothesized, inspired by earlier experiments [171,194,195,196], that cytokinesis is driven by positive feedback between contractile ring myosin and ring-directed cortical flow, with no significant resistance from cortical tension [71]. The authors showed that the per-unit-length amounts of ring myosin and anillin, ring-directed cortical flow and the constriction rate all increase with parallel exponential kinetics. In short, relaxation of the poles and membrane deposition at the polar regions would allow the cortex to be constantly pulled by myosin in the CR and ultimately loaded into the ring itself. Continuous loading of additional myosin in the CR would allow for more cortex to be pulled. This positive feedback would explain the increase in the constriction rate per unit length as the ring perimeter decreases, and consequently an overall constant constriction rate. This model implies that cortical flow and ring constriction velocity are exclusively dependent on the concentration of myosin in the CR, but no additional parameters, such as the concentration of F-actin or other crosslinkers, were evaluated.

Overall, the role of the surrounding cortex in setting cytokinesis dynamics is still a subject of debate. It is intuitive to think that the CR has to produce enough force to pinch the equator over a globally distributed actomyosin network, but experimental data of exactly when or how much the surrounding cortex resists constriction is missing outside *D. discoideum*. The contribution of cortical flows in promoting ring constriction, as proposed by Khaliulin et al. [71], remains also to be further clarified. Cortical flows have been previously proposed to be essential for ring assembly and furrow formation [61]; however, it is not totally clear whether they are a cause or a consequence of furrow ingression, or how much active material they actually load onto the constricting ring. This lack of experimental evidence is partly due to the fact that most actomyosin network components localize both in the equatorial and surrounding cortex, and, as a consequence, it becomes difficult to understand how the perturbation of a specific protein affects the mechanics of the CR or the cortex surrounding it. A challenge for the future includes the development of approaches to perturb involved proteins in a spatial and temporally controlled manner, such as rapid and reversible temperature shifts in fast-acting temperature-sensitive mutants [58], acute and localized release or uncaging of chemical compounds (reviewed in [197]), and localized optogenetic control of protein function and localization [6,198,199].

#### 4.4.2. The Importance of Estimating Mechanical and Molecular Parameters in the Same System

Current models of cytokinesis in animal cells lack experimental measurements of three key parameters: concentration of components involved, mechanical properties and turnover constants. Concentration of proteins within the contractile ring have only been measured experimentally in yeast [13,14], allowing for consistent modeling strategies to be developed in this system. Consequently, other models of cytokinesis often extrapolate protein concentrations from fission yeast, which may not be ideal when trying to obtain a quantitative description in a specific system. Besides, there are important differences between cytokinesis in yeast and other organisms, all of which may impact cytokinesis dynamics (see Section 1). Agent-based modeling of cytokinesis in animal cells will highly benefit from the implementation of a systematic strategy to measure the concentration of cytokinesis proteins in CR versus surrounding cortex over time. For example, similarly to the approach used in fission yeast [13], fluorescence intensity of fluorescently tagged proteins obtained from confocal microscopy could be calibrated for the total protein amount per cell, determined by immunoblotting. To circumvent this lack of data, most models of cytokinesis in animal cells use coarse-grained approaches, but these still require measurements of mechanical properties such as cortical tension, viscosity and flow, as well as protein dynamics, including turnover and kinetic constants for actin polymerization-depolymerization. Measurements available are limited and come primarily from fission yeast protoplasts and *D. discoideum* [60,75,153,166,200]. Mechanical parameters have also been measured in animal cells, but these have not been performed during cytokinesis [201,202]. The measurement of mechanical properties during cytokinesis is difficult because of the short window of time available. Moreover, some systems present additional complications: for instance, nematode embryos have a rigid and impermeable eggshell [203] that precludes the use of standard techniques, such as micropipette aspiration or atomic force microscopy. Turnover of cortical components, namely of actin, is also a very important feature when mechanically modeling the cell cortex. It fundamentally defines the timescale in which one can consider certain mechanical properties and flow dynamics. For instance, high actin turnover was shown to be essential to maintain constant contraction velocities and for stress generation in the CR, independently of myosin motor activity [186]. Still, the models that consider turnover dynamics as part of their assumptions are quite scarce. In particular, knowing how formin- and ARP2/3-nucleated actin filaments turnover throughout the cortex and in the CR, and how this impacts cortical thickness and mechanical properties would be of great relevance. Determining the dynamics of different actin filament pools is still challenging because formins were reported to exclude fluorescently tagged actin, presumably because of the size of GFP [13,81,204]. Efforts to quantify actin turnover at the cell equator versus poles have been made in *C. elegans* 1-cell embryos expressing GFP::actin [205]. However, the approach used did not account for myosin-dependent actin turnover and was limited to single-molecule imaging at the cell surface by TIRF microscopy, which does not allow for quantifications at the base of the ingressing furrow, where the above-mentioned incorporation problems by formins likely apply. In sum, a reliable approach to assess actin turnover in the CR during cytokinesis is still missing. Novel strategies, such as smaller fluorescent tags, or alternatively, injection of actin labelled with a small-molecule organic dye or conjugated quantum dots, could provide a necessary solution (reviewed in [206,207,208,209]).

## 5. Concluding Remarks

Given the short duration, dynamic nature and accompanying shape changes that occur during cytokinesis, its underlying mechanisms are still mysterious and will continue to fascinate biologists and physicists for the years to come. Even though progress has been substantial, some important questions remain to be tackled, especially in animal cells. How is the actomyosin network organized in the CR and surrounding cortex, what are the proteins that contribute to network connectivity in each system, how they interplay with motors to propagate tension and generate highly spatial- and temporally controlled contractility are questions still to be answered. Moreover, what the mechanical properties of cells undergoing cytokinesis are and how these are generated at the molecular level remains elusive. Coupling of experimental approaches with theoretical and computational modeling will definitely continue to bring invaluable insight into the process. Modeling will allow for possible scenarios capable of recapitulating CR formation and constriction to be built and hypotheses to be tested. In due time, a combination of agent-based and continuum approaches will eventually bring new opportunities in the modeling of cytokinesis, allowing for symbiotic recapitulations of the whole cortex with some appropriate level of microscopic detail. To test predictions from the models, the challenge will rely on advancing the experimentation in vivo and developing novel strategies that go beyond the currently available techniques. The future of research on cytokinesis is integrative and new discoveries are expected to emerge from the collective characterization of the CR and surrounding cortex biomechanics. 


**Glossary**


**Advection**—transport of a quantity (typically, a fluid) and its conserved properties by bulk motion, which is represented by a velocity field. Advection in the cell cortex happens by cortical flow and is described using Navier-Stokes equations in coarse-grained models, which are derived from the general Cauchy momentum equation for convection.

**Buckling**—deformation that occurs when a material is exposed to compressive stress, causing a deflection. F-actin buckling in the cell cortex has been attributed to compressive stresses by myosin sliding on filaments and has been implemented in agent-based models to assess its effect on network contractility.

**Cortical Tension**—force exerted on a region/piece of cortex by the cortical network around it. It is obtained by integrating the internal stresses over the thickness of the cortical layer. Cortical tension is modulated through cortex composition and organization, and cortical tension gradients lead to local contractions and cell shape changes. In coarse-grained models of cytokinesis, it is the result of both active (contraction generated by myosin motors) and viscous (intrinsic cortex deformation or flow, as well as membrane-anchoring drag forces) contributions. 

**Diffusion**—net movement of the molecules of a specimen from a region of higher concentration to a region of lower concentration. It is driven by a gradient in chemical potential of that specimen, with no bulk motion. It is commonly implemented in agent-based models by a diffusion coefficient given by the Stokes-Einstein relation for Brownian motion of spherical particles in a fluid. In continuum models, it is normally implemented as part of Reaction-Diffusion systems by means of a matrix of diffusion coefficients that changes in space and time.

**Elasticity**—property of a material that returns to its original shape after an applied stress ceases. Elastic materials are characterized by an elastic or Young modulus, which describes the resistance of the material to deformation. The Young modulus is related with the force necessary to displace an elastic material by a certain distance by the Hooke’s Law. In agent-based models, actin crosslinkers are often modelled as springs.

**Hooke’s Law**—physical law applied to elastic materials that behave as simple springs, which postulates that the force experienced by a spring is proportional to the distance displaced from the resting length (extension or compression) by the elastic or Young modulus of the spring. It is commonly used in agent-based models of the cell cortex to describe the behavior of cytoskeletal components such as actin filaments, myosin filaments and actin crosslinkers. 

**Laplace’s Law**—physical law that postulates that the tension within the surface of a spherical object under a certain internal pressure depends on its radius. It essentially describes the relationship between pressure and volume of spheres. It is mostly used in coarse-grained descriptions of the cell cortex to relate the uniform surface tension necessary to withstand a certain internal hydrostatic pressure exerted by the cytoplasm in a spherical cell with a given radius.

**Nematics**—typically used to describe a phase of liquid crystals—a state of matter with properties between conventional liquids and solid crystals—it represents a state where molecules of a material self-align with their long axis roughly parallel. Nematic alignment is a common parameter used in continuum descriptions of the cell cortex to describe the alignment of actin filaments at the equator by cortical flow.

**Probability distribution**—mathematical function that gives the probability of all the possible outcomes for a random variable in an experiment. It can be continuous or discrete, depending whether the possible scenarios for the random variable can take values in a continuous or a discrete range. The Gaussian (normal) distribution is a very common continuous probability distribution, while the Poisson distribution is a commonly used discrete probability distribution. In agent-based models of the cell cortex, probability distributions can be used to characterize, for example, the range of different actin filament lengths or orientations.

**Reaction-Diffusion systems**—mathematical models that describe the spatiotemporal change of concentration of one or more chemical specimens through processes of chemical reaction (binding/unbinding) and diffusion. Commonly used to explain the formation of travelling waves or self-emerging patterns in the cell cortex. Reaction-Diffusion kinetics of a cytoskeletal protein can be assessed experimentally by fluorescence recovery after photobleaching.

**Stiffness (or Rigidity**)—measure of the resistance of an elastic material to deformation. In agent-based models, actin filaments are sometimes viewed as rigid-rods, assuming these are straight non-flexible structures. However, some models have started including the bending ability of actin filaments, where filaments are modelled as semi-flexible strings with a certain bending rigidity.

**Viscosity**—property of a material that remains deformed even after an applied stress is removed, thus relating the viscous stress with the strain rate of that material. Viscous materials are characterized by a viscosity coefficient or fluid rigidity modulus. It essentially measures the resistance to flow, where viscous stresses are proportional to gradients of flow velocity by the viscosity coefficient. On timescales longer than the turnover of the cortex, it can be considered as purely viscous, given that elastic stresses are quickly relaxed.

**Viscoelasticity**—property of a material that shows both elastic and viscous behavior depending on the timescale stresses are applied and thus exhibiting time-dependent strain. The Maxwell (a purely viscous damper and a purely elastic spring connected in series) and Kelvin-Voigt (a Newtonian damper and Hookean elastic spring connected in parallel) models are typically used to describe this type of materials. The cell cortex is viscoelastic, behaving like an elastic solid at short time scales (typically, <1 min) and like a fluid at long time scales. 

**Young Modulus**—property that measures the stiffness of a material by the ratio between an applied stress (force over cross-sectional area) and the strain (change in length over original length), in the regime of linear elastic behavior. For a given applied stress, a material with low stiffness (lower Young Modulus) will deform more than a stiffer material (higher Young Modulus).

## Figures and Tables

**Figure 1 biomolecules-09-00194-f001:**
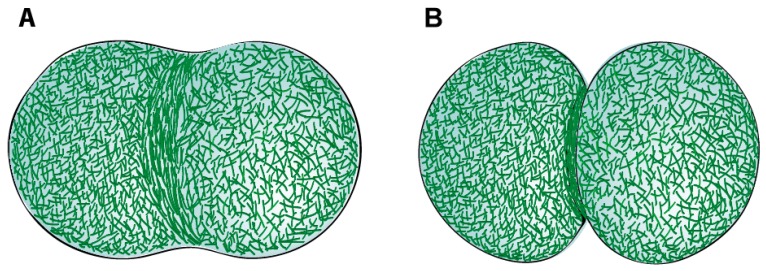
During cytokinesis, an actomyosin contractile ring (CR) specializes from and remains integrated in the cell cortex. The cartoon depicts network architectures in the contractile ring and surrounding cell cortex (**A**) at ring assembly and (**B**) during ring constriction.

**Figure 2 biomolecules-09-00194-f002:**
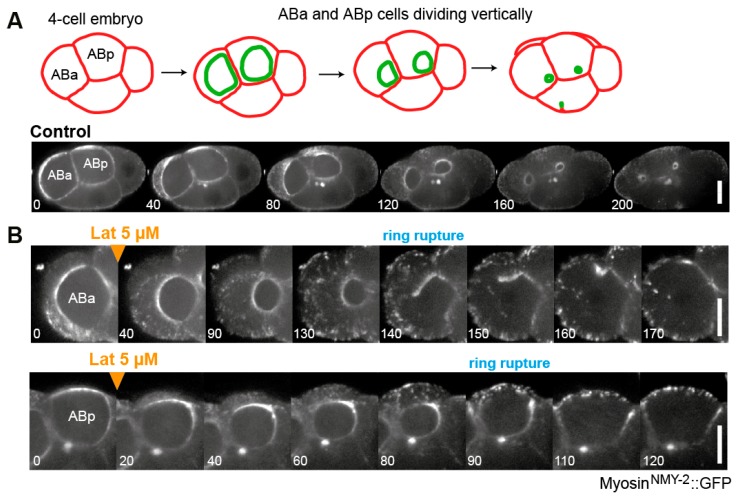
(**A**) Schematic of a 4-cell *Caenorhabditis elegans* embryo depicting the position of the ABa and ABp cells, whose division occurs perpendicularly to the imaging plane therefore providing the visualization of the entire circumference of the constricting ring (top). Stills of a timelapse video of a 4-cell embryo expressing myosinNMY-2::GFP, where both ABa and ABp rings are undergoing constriction (bottom). Time is in seconds and zero corresponds to the beginning of ring constriction. (**B**) Still images of a timelapse video of a dividing ABa (top) or ABp cell (bottom). Addition of 5 µM Latrunculin A (orange arrowheads) leads to ring rupture and cytokinesis failure in both cells. Time is in seconds and zero corresponds to frame before drug addition. Scale bars, 10 µm.

**Figure 3 biomolecules-09-00194-f003:**
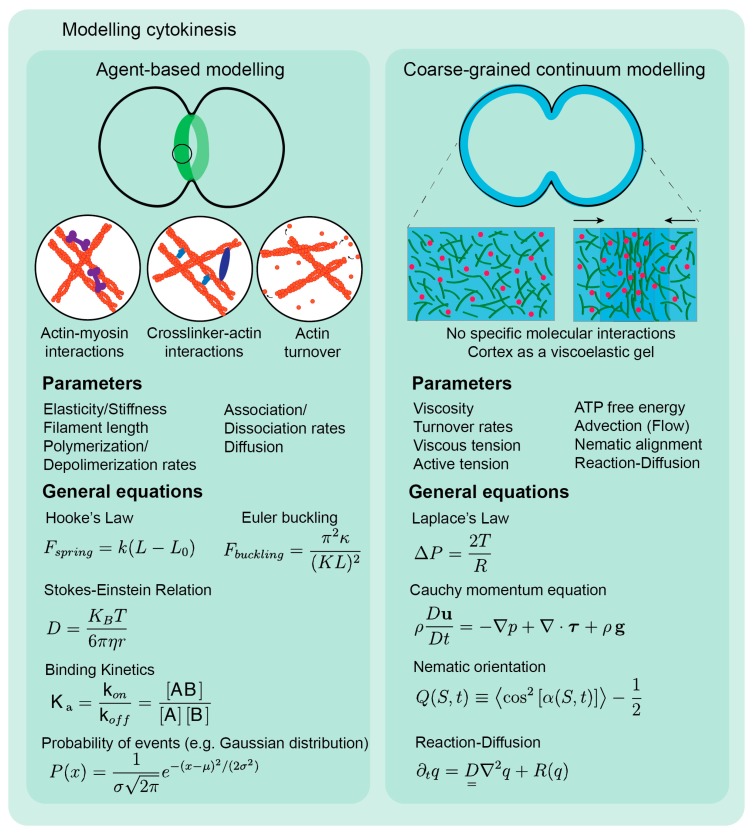
Modeling of cytokinesis is generally subdivided into two distinct approaches: agent-based simulations and coarse-grained continuum models. Agent-based models focus mostly on the CR alone and specific interactions and dynamics between cortex components are considered (myosin-actin and actin-crosslinkers interactions, as well as actin turnover dynamics). Coarse-grained models provide a description of the entire cell surface, where the cortex is seen as a viscoelastic gel able to contract in response to stress arising from physico-chemical gradients. All definitions of biochemical and mechanical parameters mentioned can be found in the Glossary, as well as their application on specific types of models of the cell cortex. The equations were chosen to convey a simplified and generalized idea of the approaches that can be used on each type of model, and therefore, may lack higher-order detail used on most mathematical models.

**Table 1 biomolecules-09-00194-t001:** Summary of the main properties of actin-targeting drugs and their effects on F-Actin networks.

Drugs	Source	Binding Site	Overall Effect on F-actin Networks	References
**Cytochalasins** (Cytochalasin D is most commonly used)	Fungi	F-actin filaments	Cap barbed ends preventing assembly and disassembly of monomers at that end. Prevent polymerization.	Reviewed in [112]
**Latrunculins** (Latrunculin A is most commonly used)	Marine sponges	Actin monomers	Prevent polymerization by sequestering monomers, increase depolymerization at both filament ends and promote filament severing	[113,114,115]
**Jasplakinolide**	Marine sponges	F-actin filaments	Stabilizes F-actin filaments, induces polymerization of actin into amorphous masses	[116,117,118]
**Phalloidin**	Fungi	F-actin filaments	Stabilizes F-actin filaments	Reviewed in [112]

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
