# Peer review of "Network Contractility during Cytokinesis—From Molecular to Global Views"

_biomolecules, 2019, doi:10.3390/biom9050194_

Round 1
Reviewer 1 Report
Leite Carvalho Biomolecules 4-19
This is a timely, authoritative review of research on cytokinesis. Better than many reviews, the authors are appropriately critical of the limitations of the published work. Therefore, I am enthusiastic about publishing the review, but I have many suggestions for improving this version.
L 35: reword “the base of the ingressing equatorial furrow.”
L 89: “The linear actin filaments of the CR are nucleated and/or elongated by formins of the diaphanous family (reviewed in [33]).” The identification of the participating formins in animal cells is still uncertain. For example, the papers by the Watanabe group first showed that mDia2 appeared to be the relevant formin in cultured cells. But when they knocked out mDia2 in mice, the embryos went through many rounds of cell division before cytokinesis failed in blood cells. The situation in C. elegans is simpler because of fewer formin isoforms, but even there it is not yet clear whether formins besides CYK-1 participate.
L 96: reword “…is a barbed‐end directed actin motor…”
L 99: reword “…composed of two heavy chain and two copies of two different light chains…”
L 102: reword “…heads located at both ends…” and “These heads bind actin so a bipolar myosin filaments can crosslink and drive the sliding of anti-parallel actin filaments.”
L 131: reword “…the CR maintains its thickness as it constricts, which implies that the actin filaments disassemble as its circumference decreases.”
L 262: The interpretation of these experiments with mutant forms of myosin-II might also consider contributions from other myosin isoforms.
L 287: Electron microscopy of thin sections is not an accurate method to measure the lengths of actin filaments, so references 27 and 75 should not be cited as evidence supporting “actin filaments become shorter during constriction.”
L 292: Latrunculin. I think that the evidence for “increase depolymerization at both filament ends and promote filament severing” is from a recent paper by Fujiwara.
Pages 7-8: The extent of the coverage of experiments on drug treatments may be excessive, given that many of them are hard to interpret and therefore not that informative about mechanisms.
Paragraph beginning on line 324 has a number of statements that should be clarified. What is the evidence in ref 35 that CYK-1 is active during ring constriction? The statement “the cytokinesis formin Cdc12p in fission yeast may be inactive during ring constriction because it is inhibited by myosin pulling on formin‐ bound actin filaments during ring assembly [114]” is misleading. The Zimmerman paper showed that Myo2 reduced the rate of actin filament elongation but did not stop elongation. I do not understand how the cofilin experiments provide evidence regarding actin polymerization during ring constriction. The experiments with fission yeast ghosts ± drugs are not informative about whether actin polymerization occurs during constriction in live cells, because the resistance to constriction is far less in ghosts without turgor pressure. The modeling paper by Stachowiak et al. (Devel Cell 2014) shows why turnover of the actin filaments and myosin is required mechanically to maintain tension.
L 351: What is the evidence that anillin or septins are actin filament crosslinkers?
L 352: Reconsider “molecule flexibility” in the following: “Depending on their size and structural properties, like molecule flexibility, these crosslinkers lead to the formation of actin filament networks with distinct morphologies and stiffness levels.” Decades ago Fred Lanni and other labs showed that the formation of bundles vs. isotropic networks depends on the concentrations of the filaments, the length of the filaments, concentration of the crosslinkers and the affinity (really the dissociation rate constant) of the crosslinkers for the filaments.
L 372: The effects of overexpressing crosslinkers is not informative about the normal functions of these proteins.
L 396: Is myosin-VI really “processive and double‐ headed?” The reader should probably be told that the odd thing about the experiment in Ref 138 is that myosin-VI is a pointed end motor.
L 404: In the following “Other in silico models” the simulations not the models are in silico.
L 406: The following “Nonetheless, whether ordered/modular anti parallel filaments or disordered parallel filaments best reflect the cytokinetic CR” does not include all of the options. Disordered anti-parallel filaments may be the most likely option.
L 412: Reword “Intermediate levels of myosin-II (NMY‐ 2) result in a ….”
L 415: I do not understand the “reason why cytokinesis would not be optimized for maximal speed is enigmatic.” Many reasons exist including constricting at a rate that allows the expansion of the plasma membrane in the furrow.
L 444: I do not understand “The limitation of using agent‐ based modelling to understand the mechanics of actomyosin networks is that it is difficult to know how each component contributes to set the network mechanical properties.” This is not difficult, but requires serious biochemical and biophysical characterization of the components for integration into molecularly-explicit models. Obviously, this is better than continuum models where the assumptions are vague. Therefore, I disagree that continuum models are “more reliable.”
L 446: The following is misleading “agent‐ based numerical simulations are computationally unworkable when modelling large‐ scale systems such as the entire cell cortex.” One does not have to model a whole cell, just a small part of the cortex or contractile ring suffice.
Paragraph starting on L 515: You might note that a general failing of the network models is that they do not include anchoring the barbed ends of the actin filaments, which is expected given formins at their barbed ends and which is the key to getting tension from the system.
L 565: The premise of this paragraph is unfair. The failure to observe nodes in reconstructions of cryo-EM tomograms cannot negate the overwhelming evidence for nodes from observations on live cells. Negative results do not trump positive evidence. More questionable are the assumptions for the models in ref 173. The modeling itself also has some problems, although expertise in modeling is required to recognize those shortcomings.
L 578-632: The future directions section has lots of history, which might be better in the previous section.
L 581: reword “…disregard the fact that the CR is a differentiated part of the cortex in animal cells...”
L 596: reword “….proposed that resistive forces from the surrounding cortex counteract actively generated contractile tension at the cell equator…
L 674: The following is an understatement: “A totally reliable approach to assess actin turnover during cytokinesis is still missing.” In fact, since GFP-actin is not fully functional and no one has done measurements on contractile rings in cells injected with actin labeled with a fluorescent dye, no reliable data are available on actin turnover in any cell including fission yeast. I am very skeptical about the proposal to use nanobodies, which will only measure the turnover the nanobodies, not the actin.
L 681: reword “Even though progress has been substantial…”
L 692: reword “The future of research on cytokinesis is integrative…”
I hope that these suggestions will be helpful.
Author Response
Response to Reviewer 1 Comments
This is a timely, authoritative review of research on cytokinesis. Better than many reviews, the authors are appropriately critical of the limitations of the published work. Therefore, I am enthusiastic about publishing the review, but I have many suggestions for improving this version.
L 35: reword “the base of the ingressing equatorial furrow.”
Re-worded.
L 89: “The linear actin filaments of the CR are nucleated and/or elongated by formins of the diaphanous family (reviewed in [33]).” The identification of the participating formins in animal cells is still uncertain. For example, the papers by the Watanabe group first showed that mDia2 appeared to be the relevant formin in cultured cells. But when they knocked out mDia2 in mice, the embryos went through many rounds of cell division before cytokinesis failed in blood cells. The situation in C. elegans is simpler because of fewer formin isoforms, but even there it is not yet clear whether formins besides CYK-1 participate.
We agree that the identification of the participating formins has not been thoroughly studied in some systems. We now refer to formins in general, without specifying a specific family.
L 96: reword “…is a barbed‐end directed actin motor…”
Re-worded.
L 99: reword “…composed of two heavy chain and two copies of two different light chains…”
Re-worded.
L 102: reword “…heads located at both ends…” and “These heads bind actin so a bipolar myosin filaments can crosslink and drive the sliding of anti-parallel actin filaments.”
Re-worded.
L 131: reword “…the CR maintains its thickness as it constricts, which implies that the actin filaments disassemble as its circumference decreases.”
Re-worded.
L 262: The interpretation of these experiments with mutant forms of myosin-II might also consider contributions from other myosin isoforms.
We included this possibility in the end of section 3.2.
L 287: Electron microscopy of thin sections is not an accurate method to measure the lengths of actin filaments, so references 27 and 75 should not be cited as evidence supporting “actin filaments become shorter during constriction.”
We now say that measurement of actin filament length by transmission EM of thin sections indicates that actin depolymerization is required for cytokinesis.
L 292: Latrunculin. I think that the evidence for “increase depolymerization at both filament ends and promote filament severing” is from a recent paper by Fujiwara.
We apologize for the mistake and corrected the reference in Table 1.
Pages 7-8: The extent of the coverage of experiments on drug treatments may be excessive, given that many of them are hard to interpret and therefore not that informative about mechanisms.
We decreased it a bit but kept most of the information as we believe that a thorough compilation of the experiments done in several systems may be useful to the reader.
Paragraph beginning on line 324 has a number of statements that should be clarified. What is the evidence in ref 35 that CYK-1 is active during ring constriction? The statement “the cytokinesis formin Cdc12p in fission yeast may be inactive during ring constriction because it is inhibited by myosin pulling on formin‐ bound actin filaments during ring assembly [114]” is misleading. The Zimmerman paper showed that Myo2 reduced the rate of actin filament elongation but did not stop elongation. I do not understand how the cofilin experiments provide evidence regarding actin polymerization during ring constriction. The experiments with fission yeast ghosts ± drugs are not informative about whether actin polymerization occurs during constriction in live cells, because the resistance to constriction is far less in ghosts without turgor pressure. The modeling paper by Stachowiak et al. (Devel Cell 2014) shows why turnover of the actin filaments and myosin is required mechanically to maintain tension.
1) We explained that inactivation of CYK-1 during constriction was found to substantially decrease the rate of constriction
2) We agree that the Zimmerman paper showed a decrease of the elongation rate by myosin pulling for Cdc12 but not for mDia2, and therefore removed that sentence and reference
3) We realize that our text was confusing. We have now reorganized section 3.4. to clearly separate data on actin polymerization and depolymerization. The data on cofilin provides evidence regarding the importance of actin depolymerization.
4) We agree that experiments with fission yeast ghosts are not so informative. However, it is our opinion that that evidence together with experiments in intact yeast and yeast spheroplasts using drugs do indicate that actin turnover may not be required for ring contraction per se, but rather to maintain actin filament homeostasis during the process.
5) We included the modelling paper from Stachowiak et al.
L 351: What is the evidence that anillin or septins are actin filament crosslinkers?
We added references in the text evidencing the actin filament crosslinking capability of all mentioned passive actin filament crosslinkers. More specifically, anillin was shown to bundle/crosslink actin filaments in vitro in the works of Kinoshita et al., 2002 and Jananji et al., 2017. Septins were shown to bundle/crosslink actin filaments in vitro in the works of Kinoshita et al., 2002 and Mavrakis et al., 2014.
L 352: Reconsider “molecule flexibility” in the following: “Depending on their size and structural properties, like molecule flexibility, these crosslinkers lead to the formation of actin filament networks with distinct morphologies and stiffness levels.” Decades ago Fred Lanni and other labs showed that the formation of bundles vs. isotropic networks depends on the concentrations of the filaments, the length of the filaments, concentration of the crosslinkers and the affinity (really the dissociation rate constant) of the crosslinkers for the filaments.
We agree with the reviewer’s suggestion and made alterations accordingly.
L 372: The effects of overexpressing crosslinkers is not informative about the normal functions of these proteins.
We agree and have added this observation to the sentence
L 396: Is myosin-VI really “processive and double‐ headed?” The reader should probably be told that the odd thing about the experiment in Ref 138 is that myosin-VI is a pointed end motor.
This is an important piece of information that we have included. We also expanded the paragraph in order to clarify the results pertaining in vitro micropatterned contractile rings.
L 404: In the following “Other in silico models” the simulations not the models are in silico.
We totally agree and corrected this.
L 406: The following “Nonetheless, whether ordered/modular anti parallel filaments or disordered parallel filaments best reflect the cytokinetic CR” does not include all of the options. Disordered anti-parallel filaments may be the most likely option.
We agree and included this option in the sentence.
L 412: Reword “Intermediate levels of myosin-II (NMY‐ 2) result in a ….”
Re-worded.
L 415: I do not understand the “reason why cytokinesis would not be optimized for maximal speed is enigmatic.” Many reasons exist including constricting at a rate that allows the expansion of the plasma membrane in the furrow.
We agree that several reasons may exist but we still think it is enigmatic. We re-wrote the sentence to convey our thoughts better. We think that “faster ring constriction comes with a price for the cell: since crosslinkers are also located outside the contractile ring, faster constriction as a result of decreased levels of crosslinkers could lead to decreased cortical tension, and this could compromise the integrity of the cell.”
L 444: I do not understand “The limitation of using agent‐ based modelling to understand the mechanics of actomyosin networks is that it is difficult to know how each component contributes to set the network mechanical properties.” This is not difficult, but requires serious biochemical and biophysical characterization of the components for integration into molecularly-explicit models. Obviously, this is better than continuum models where the assumptions are vague. Therefore, I disagree that continuum models are “more reliable.”
We agree it was not totally explicit why it is “difficult” and rephrased the sentence to better convey the message. In our opinion, the assumptions in continuum models are not vague, since these are based on demonstrated physical laws and variables have physical meaning. Of course, there’s the problem that variables such as viscosity and cortical tension need to be measured in vivo. Agent-based models are very interesting and useful to address questions involving specific interactions between components at the microscopic level and how this in turn impacts the macroscopic behavior of the network. However, because of the lack of biochemical and biophysical characterization of the biological organisms, there are situations where a lot of assumptions need to be made. We understand the reviewer’s point and agree that the reliability of a specific type of model over the other will always depend on the question itself. Therefore, we tried to reformulate the text in order to better explain when continuum models can be “more adequate” than agent-based ones. Additionally, we totally agree that a symbiosis between the two types of models may actually be more enlightening in the future and included this in the concluding remarks.
L 446: The following is misleading “agent‐ based numerical simulations are computationally unworkable when modelling large‐ scale systems such as the entire cell cortex.” One does not have to model a whole cell, just a small part of the cortex or contractile ring suffice.
We agree with the reviewer that the sentence could be misleading. We agree that there’s no need to model the whole cell, but in our opinion, this would depend on the question. For example, if one is trying to understand how contractility happens in the contractile ring and in the cortex outside the ring, where actin networks show different architectures (linear vs. branched), then modelling just a small part of the contractile ring or the remaining cortex would be enough. On the other hand, if one is investigating how the overall distribution of a component (e.g. myosin-II) over the cell surface affects the ring positioning and constriction velocity, then it is necessary to model the entire cortex, which for the moment is unrealistic using agent-based models. To make it more clear, we rephrased this sentence.
Paragraph starting on L 515: You might note that a general failing of the network models is that they do not include anchoring the barbed ends of the actin filaments, which is expected given formins at their barbed ends and which is the key to getting tension from the system.
This is a very good point that we now included.
L 565: The premise of this paragraph is unfair. The failure to observe nodes in reconstructions of cryo-EM tomograms cannot negate the overwhelming evidence for nodes from observations on live cells. Negative results do not trump positive evidence. More questionable are the assumptions for the models in ref 173. The modeling itself also has some problems, although expertise in modeling is required to recognize those shortcomings.
We agree that it seemed we were questioning the existence of nodes ourselves. That is not what we meant and therefore we rephrased the sentence and also included some comments about the assumptions made in the model.
L 578-632: The future directions section has lots of history, which might be better in the previous section.
We understand this point but as the interplay between the ring and surrounding cortex is still a subject of debate and there's not much experimental evidence for it, we wanted to separate this from the rest. As research in this topic is an ongoing challenge, we think it belongs in the section 4.4, which we now renamed “Open questions and future directions on cytokinesis modelling”.
L 581: reword “…disregard the fact that the CR is a differentiated part of the cortex in animal cells...”
Re-worded.
L 596: reword “….proposed that resistive forces from the surrounding cortex counteract actively generated contractile tension at the cell equator…
Re-worded.
L 674: The following is an understatement: “A totally reliable approach to assess actin turnover during cytokinesis is still missing.” In fact, since GFP-actin is not fully functional and no one has done measurements on contractile rings in cells injected with actin labeled with a fluorescent dye, no reliable data are available on actin turnover in any cell including fission yeast. I am very skeptical about the proposal to use nanobodies, which will only measure the turnover the nanobodies, not the actin.
We agree and have re-written the text accordingly.
L 681: reword “Even though progress has been substantial…”
Re-worded.
L 692: reword “The future of research on cytokinesis is integrative…”
Re-worded.
I hope that these suggestions will be helpful.
They were extremely helpful. We would like to thank the Reviewer for the thorough reading and for significantly improving our review.
Reviewer 2 Report
Leite et al have written an incredibly comprehensive review of cytokinesis, delving into cytokinetic ring architecture, the roles of myosin II and crosslinkers, and the benefits models provide to advance the field forward. The authors nicely integrate and compare knowledge across multiple systems, citing the historical progression of the field as well as more recent developments. The authors should include a section on the mechanical and material properties of the cortex. The authors describe the relative tensions in the equatorial region versus the polar cortex, but a section considering viscoelasticity, cortical tension, and the material properties as they change during cytokinesis would further develop the “global views” part of the manuscript. While terms are included in the modeling cytokinesis figure panel 3, definitions and descriptions of those terms are not explicitly included (viscoelasticity, tension, Laplace pressure, Nematic alignment, etc.). A reader might benefit from having the definitions of these as well as commentary to help understand how mechanics influences each of the components described here. My further comments here are simply suggestions to provide further evidence to support the author’s statements or to clarify certain statements, both with the intent of strengthening this already fantastic review.
- The authors discuss differences between cytokinesis in yeast versus other systems. They might add the other critical point here that actin concentrations and filament distributions are significantly different in yeast than in other systems (see ref 53, Wu and Pollard, Science 2005, and Robinson et al, Comp Biophys Rev 2012). Additionally, protein concentration differences likely play key roles as well (crosslinkers, small GTPases, etc – again Wu and Pollard, Descovich et al, Mol Biol Cell 2018, and Kothari et al, JCS 2019).
- While describing myosin II filaments in section 2.2, perhaps it would also be useful to mention a recent study that describes the pathway of myosin II assembly as this adds to our knowledge of the effects of phosphorylation on activation of myosin II (Liu, Shu, and Korn, PNAS 2018).
- While EM was recently done in sea urchin embryos, it may also be worth noting that inDictyostelium, EM studies show the “ring” is in fact a dense meshwork of actin filaments (Reichl, et al Curr Biol, 2008). Comparing these EM structures and the staining in HeLa cells (53) to the structures in the embryos could highlight differences between different size-scale cells.
- (Section 4.3) The authors very nicely describe the forces and tension differential between the equatorial region and the polar cortex. In fact this differential plays a key role in myosin II accumulation to the furrow (Ren et al, Curr Biol 2009) and may be mentioned around lines 186-188.
- Excellent analysis of myosin II mutants. Section 3.3 could include specific mechanisms. For example, cells undergoing myosin-independent cytokinesis use Laplace-mediated pressures and the properties/ geometries of the material to complete cytokinesis (modeling and analysis in Zhang and Robinson, PNAS 2005, also likely true in ref 92, 93).
- As this review is of interest to a broad audiences, please include descriptions of “ABa and ABb” in Fig. 2, or refer to them as cell 1 and 2.
- Please reorganize section 3.4. Content is great, but authors switch from polymerization to depolymerization within the section. Describe the importance of one, and then switch to the opposite.
- The authors are correct in describing that crosslinkers can localize to the furrow as well as to the polar cortex. Many of these protein concentrations have been measured in Dictyostelium in equatorial:polar cortex regions, which might be worth including (ref 84, Reichl et al Curr Biol 2008, reviewed in Robinson et al, Comp Biophys Rev 2005).
- Correction to lines 382-383: In fact myosin and cortexillin retain their cooperative accumulation during interphase as well. Later studies demonstrated that different thresholds due to pressure differentials were required to measure the co-dependence of the interaction (Luo et al, Nat Mat 2013). The pressure differential was required to deshield the myosin / cortexillin to allow them to sense the load on the system.
- Authors do a great job of reviewing overall modeling of actomyosin networks and cytokinesis in section 4. A little more detail on specific models might be helpful. I refer authors to Poirier et al Plos Comp Bio 2012 Mohan et al J R Soc Interface 2015 to see examples of such models of cytokinesis.
- Phrasing of lines 613-625 might be reconsidered. Phrased as is, it appears the authors claim that resistive or frictional cortical forces are separate or mutually exclusive from the cortical flow that also contributes to myosin and anillin accumulation.
Author Response
Response to Reviewer 2 Comments
Leite et al have written an incredibly comprehensive review of cytokinesis, delving into cytokinetic ring architecture, the roles of myosin II and crosslinkers, and the benefits models provide to advance the field forward. The authors nicely integrate and compare knowledge across multiple systems, citing the historical progression of the field as well as more recent developments. The authors should include a section on the mechanical and material properties of the cortex. The authors describe the relative tensions in the equatorial region versus the polar cortex, but a section considering viscoelasticity, cortical tension, and the material properties as they change during cytokinesis would further develop the “global views” part of the manuscript. While terms are included in the modeling cytokinesis figure panel 3, definitions and descriptions of those terms are not explicitly included (viscoelasticity, tension, Laplace pressure, Nematic alignment, etc.). A reader might benefit from having the definitions of these as well as commentary to help understand how mechanics influences each of the components described here. My further comments here are simply suggestions to provide further evidence to support the author’s statements or to clarify certain statements, both with the intent of strengthening this already fantastic review.
We thank the Reviewer for being enthusiastic about our review and for all the enriching suggestions.
We totally agree with the reviewer that clarifying the mentioned concepts is a necessary improvement. Therefore, we are now including a glossary with the concepts and physical laws that we think needed clarification. Instead of writing a new section exclusively on the mechanical properties of the cortex, which we thought does not fit so well with the flow of the text, we include in the glossary specific parameters used in the models and how they relate with specific mechanical properties of the cortex.
- The authors discuss differences between cytokinesis in yeast versus other systems. They might add the other critical point here that actin concentrations and filament distributions are significantly different in yeast than in other systems (see ref 53, Wu and Pollard, Science 2005, and Robinson et al, Comp Biophys Rev 2012). Additionally, protein concentration differences likely play key roles as well (crosslinkers, small GTPases, etc – again Wu and Pollard, Descovich et al, Mol Biol Cell 2018, and Kothari et al, JCS 2019).
We included the differences in filament distribution in section 3.1 (Actomyosin structure of the contractile ring). We thought about adding the differences in concentrations but decided not to because, to our knowledge, there is little data available that can be compared. Comparable values that we could find are of actin and myosin concentrations in rings of fission yeast and D. discoideum: Dicty 6.4 µM myosin, 80-150 µM actin; S. pombe: 20 µM myosin (Myo2p) and 460 µM actin (approximately 3-fold more actin and myosin in fission yeast than in Dicty).
- While describing myosin II filaments in section 2.2, perhaps it would also be useful to mention a recent study that describes the pathway of myosin II assembly as this adds to our knowledge of the effects of phosphorylation on activation of myosin II (Liu, Shu, and Korn, PNAS 2018).
We acknowledge the importance of this data and therefore not only included it but also improved the paragraph concerning myosin polymerization in section 2.2
- While EM was recently done in sea urchin embryos, it may also be worth noting that in Dictyostelium, EM studies show the “ring” is in fact a dense meshwork of actin filaments (Reichl, et al Curr Biol, 2008). Comparing these EM structures and the staining in HeLa cells (53) to the structures in the embryos could highlight differences between different size-scale cells.
We welcome the suggestion and added a new sentence on this subject in section 3.1.
- (Section 4.3) The authors very nicely describe the forces and tension differential between the equatorial region and the polar cortex. In fact this differential plays a key role in myosin II accumulation to the furrow (Ren et al, Curr Biol 2009) and may be mentioned around lines 186-188.
In that part of the text, we are referring to recruitment of myosin filaments to the cell equator and therefore we now include the mentioned reference to explain that a mechanosensing-mediated mechanism has also been proposed to lead to cooperative myosin accumulation at the furrowing cortex.
- Excellent analysis of myosin II mutants. Section 3.3 could include specific mechanisms. For example, cells undergoing myosin-independent cytokinesis use Laplace-mediated pressures and the properties/ geometries of the material to complete cytokinesis (modeling and analysis in Zhang and Robinson, PNAS 2005, also likely true in ref 92, 93).
We agree and incorporated these specific mechanisms in section 3.3
- As this review is of interest to a broad audiences, please include descriptions of “ABa and ABb” in Fig. 2, or refer to them as cell 1 and 2.
We expanded figure 2 legend in order to better explain the organization of the 4 cell embryo
- Please reorganize section 3.4. Content is great, but authors switch from polymerization to depolymerization within the section. Describe the importance of one, and then switch to the opposite.
We fully agree this section needed to be reorganized and did as requested.
- The authors are correct in describing that crosslinkers can localize to the furrow as well as to the polar cortex. Many of these protein concentrations have been measured in Dictyostelium in equatorial:polar cortex regions, which might be worth including (ref 84, Reichl et al Curr Biol 2008, reviewed in Robinson et al, Comp Biophys Rev 2005).
We included this reference in the context of section 4.4.2 (The importance of estimating mechanical and molecular parameters in the same system) to convey the message that measurements available are limited and come primarily from fission yeast and D. discoideum. We also included it in section 3.5 (Involvement of actin filament crosslinkers in cytokinesis) when we referred to different crosslinker dynamics in equatorial and polar regions. However, we thought that adding the concentrations of crosslinkers in equatorial and polar regions would not fit the current flow of the text in section 3.5.
- Correction to lines 382-383: In fact myosin and cortexillin retain their cooperative accumulation during interphase as well. Later studies demonstrated that different thresholds due to pressure differentials were required to measure the co-dependence of the interaction (Luo et al, Nat Mat 2013). The pressure differential was required to deshield the myosin / cortexillin to allow them to sense the load on the system.
We thank the reviewer for the clarification. However, as our text is focused on cytokinesis, we removed the mention to interphase.
- Authors do a great job of reviewing overall modeling of actomyosin networks and cytokinesis in section 4. A little more detail on specific models might be helpful. I refer authors to Poirier et al Plos Comp Bio 2012 Mohan et al J R Soc Interface 2015 to see examples of such models of cytokinesis.
We thank the reviewer for the suggestions and indeed included Poirier et al., 2012, an important study that was missing. We decided however not to include Mohan et al J R Soc Interface 2015, for the mere reason that, even though this study is very interesting in what regards mechanosensing abilities of myosin and how this affects contractility in different biological processes, it does not directly explore the cell deformation occurring in cytokinesis, but rather that occurring during micropipette aspiration. Since our modelling section is exclusively dedicated to models of cytokinesis, we thought this study would not fit in.
- Phrasing of lines 613-625 might be reconsidered. Phrased as is, it appears the authors claim that resistive or frictional cortical forces are separate or mutually exclusive from the cortical flow that also contributes to myosin and anillin accumulation.
We agree and rephrased it to better convey the message.
Reviewer 3 Report
Manuscript "Network contractility during cytokinesis ‐ from
3 molecular to global views" by Leite and co-authors reviews current state of knowledge about contractile ring functions. The manuscript is nicely written and contains a log of useful information. In particular I liked the very detailed insights into history and on-going efforts in the modelling of contractile ring. This work would be very useful for expertise in the field and also for cell biology students.
Minor comments
Line 185. I suggest changing "earlier proposals" for "earlier theories".
Author Response
Manuscript "Network contractility during cytokinesis ‐ from molecular to global views" by Leite and co-authors reviews current state of knowledge about contractile ring functions. The manuscript is nicely written and contains a log of useful information. In particular I liked the very detailed insights into history and on-going efforts in the modelling of contractile ring. This work would be very useful for expertise in the field and also for cell biology students.
We thank the Reviewer for the kind analysis of our review.
Minor comments
Line 185. I suggest changing "earlier proposals" for "earlier theories".
This sentence has been re-written.